# Crystallization of Form II Paracetamol with the Assistance of Carboxylic Acids toward Batch and Continuous Processes

**DOI:** 10.3390/pharmaceutics14051099

**Published:** 2022-05-20

**Authors:** Kuan-Lin Yeh, Hung-Lin Lee, Tu Lee

**Affiliations:** Department of Chemical and Materials Engineering, National Central University, 300 Zhongda Road, Zhongli District, Taoyuan City 320317, Taiwan; xp871115@yahoo.com.tw (K.-L.Y.); sky770703@hotmail.com (H.-L.L.)

**Keywords:** oxalic acid, fumaric acid, seeding, solution complex, purification

## Abstract

Form II paracetamol has captured the interest of researchers due to its improved compressibility. However, its low stability has made it difficult to be produced on a large scale with good reproducibility. In the present study, the selective polymorphic formation of paracetamol was carried out by cooling crystallization with four types of additives: adipic acid, fumaric acid, oxalic acid, and succinic acid. It was found that: (1) the more additives that were added, the higher the probability of forming Form II paracetamol; (2) Form II paracetamol could be induced by seeding the paracetamol aqueous solution with Form II paracetamol and fumaric acid crystals, and not the other three carboxylic acids; (3) a new solution complex of paracetamol–oxalic acid, evidenced by the solubility diagram, was responsible for the selective nucleation of Form II paracetamol in the oxalic acid aqueous solution; and (4) the range of the degree of supersaturation for nucleating Form II paracetamol was extended with the assistance of oxalic acid or fumaric acid. In large-scale crystallization, Form II paracetamol was produced by the continuous crystallization of 44 mg of paracetamol/mL in 50 wt% of fumaric acid aqueous solution with a flow rate of 150 mL/min.

## 1. Introduction

Polymorphism is a common phenomenon in which a substance exists in multiple ordered structures with different packing or conformational arrangements [1,2]. The diversity in molecular arrangements gives rise to various physicochemical properties, such as solubility, dissolution rate, stability [3], hygroscopicity [4], morphology [5], color [6], and taste [7]. Those properties are associated with the process and product performance. Polymorphism provides materials with improved properties without changing their original chemical entities. Chocolate is one of the famous examples of a commercial polymorphic product. Cocoa butter, the major ingredient in chocolate, can exist in six polymorphs (Forms I to VI) with different textures and melting points [8,9]. Form V, which is glassy, firm, and good to snap, possesses the best taste, while the other polymorphs are too soft, easily crumbled and melted, and hard to snap. As a patentable object, new polymorphs of an active pharmaceutical ingredient (API), as well as its chemical entity, are generally patented to prolong the market exclusivity [10]. By a rough estimate, about half of APIs or drug substances possess multiple polymorphs [11]. However, the inevitable polymorphic transformation during manufacturing or storage causes a huge challenge in its application in the pharmaceutical industry [12].

Paracetamol (PCA), also known as acetaminophen, is one of the most commonly used antipyretic and analgesic APIs in many prescriptions and over-the-counter drugs. Nowadays, it is an essential medicine to keep at home around the world. The demand for first-line antipyretics including PCA has considerably increased to relieve the side effects of vaccines combating the COVID-19 pandemic. Most of the commercially available drug products of PCA are in oral solid dosage forms (i.e., tablets and capsules), while other dosage forms such as intravenous injection and rectal suppository are relatively rare. For the solid dosage forms, the polymorphic control of PCA is crucial to its structural purity and properties. 

To date, nine anhydrous polymorphs of PCA had been discovered and named Forms I to IX. Their phase transitions are illustrated in Figure 1. Forms I, II, and III PCA were accessible by solution crystallization under ambient temperature and pressure [13,14,15]. Forms IV and V were produced by compressing PCA at high pressure (Figure 1) [16]. Form VI was formed from Form III by cooling to between 170 and 220 K [17]. Forms VII, VIII, and IX were obtained by melt crystallization on a hot stage at 50 to 80 °C [18]. Except for Forms I, II, and III PCA, the other polymorphs were found in recent years, and thus their crystallographic data, stability, and physicochemical properties are lacking. In addition, three hydrate forms of mono- [19], di- [20], and tri-hydrate [21] and seven solvates of PCA were reported as well [22,23,24].

Dry metastable Form II PCA crystals were unchanged without the polymorphic transformation to the stable Form I upon storage at room temperature for several months [25]. If Form II PCA was contaminated with Form I or exposed to high moisture or underwent mechanical grinding, the transformation from Form II to Form I could be accelerated [26]. Form III PCA, less stable than Form II, was easily transformed into Forms II and I in air [27]. Although Forms II and III showed the advantages of bioavailability and compressibility, their lower stabilities have made the preparation with high polymorphic purity and reproducibility very difficult. This is especially so for Form III PCA.

A variety of crystallization methods had been developed to prepare metastable Form II PCA, such as melting, cooling or evaporation [28,29,30,31], seeding [32,33,34,35], contact line [36], the use of foreign substrates [37,38,39,40], ultrasound [41,42], reaction coupling [43], and multicomponent addition [44,45,46,47]. However, some challenges still remain to be addressed and overcome, as follows [48]. Melt crystallization is commonly used to discover and prepare different polymorphs in the laboratory, while solution crystallization is preferred and more realistic in pharmaceutical manufacturing. On creating a high degree of supersaturation (such as by rapid cooling), the highly energetic Form II PCA could be induced, but then Form I nuclei might be generated at a lower degree of supersaturation. The selective formation of Form II PCA was carried out on a gram scale by seeding Form II crystals at −10 to 0 °C [33]. Such requirements for the rigorous control of crystallization kinetics of Form II PCA had subjected those crystallization processes to insufficient stability. Contact line and ultrasound-assisted crystallization methods were restricted to a small scale and very specific equipment. The use of a heterogeneous particle or a foreign substrate may direct the crystallization of PCA to a certain polymorph, but an extra purification step was necessary to isolate it from the desired polymorphic product. Other special technologies, such as the combination of metal-assisted and microwave-accelerated evaporation [49], inkjet printing [50], plasmonic trapping-induced crystallization [51], electrical confinement [52], and laser irradiation [53] have been reported, however, all of them are seldom employed in manufacturing.

Reaction coupling and multicomponent crystallization seem to be the most reliable methods that can be employed for high reproducibility and feasibility for scaling up. Although the combination of chemical synthesis and subsequent crystallization steps makes “reaction coupling” close to a practical process, the temporary loss of control during the holding time without agitation could be disadvantageous [43]. Multicomponent crystallization routes have been promoted to control the polymorphic formation with the assistance of certain additives. Benzoic acid, 4-halobenzoic acid, and metacetamol were added to control the formation of Form II PCA through evaporation or cooling crystallization [44,45]. Agnew et al. [45] and Nicoud et al. [47] developed continuous crystallization with the use of metacetamol for making Form II PCA. The similar molecular structure and interaction between PCA and metacetamol caused metacetamol not only to adsorb onto the crystal surface but also to be incorporated into the crystal lattice of PCA, thus suppressing the formation of Form I PCA. Even worse, the remaining metacetamol reduced the chemical (compositional) purity [54], and its removal by a separate recrystallization step might cause the polymorphic transformation of Form II PCA [55].

Although the effects of structure-related or tailor-made additives on solubility, crystal habit, polymorphism, and enantiomeric purity have often been interpreted, the exact prediction of which additive is able to control the desirable polymorphic formation is still vague and intractable. It is quite clueless and complicated to ferret out a proper additive from a large pool of derivatives of the desired products. The similar chemical structures between the additive and the product increase the difficulty in characterization and purification as well. It can be anticipated that an additive having a similar chemical structure to the product usually has a solubility close to the one for the product. In our previous work, two co-formers, oxalic acid (OXA) and maleic acid (MAL), give a similar structural arrangement of PCA in the co-crystals to that of Form II PCA, which could offer a guideline for selecting the additive candidates [56]. Intriguingly, Form II PCA was stably produced along with the precipitation of fumaric acid (FUM) as an impurity converted from the isomerization of MAL in the aqueous solution of MAL, but some batches yielded Form I PCA when FUM failed to precipitate out [56]. Although OXA and MAL could selectively induce the Form II PCA formation, the mechanism and the effects such as the type of additives, the weight ratio of PCA to additive, the initial concentration (i.e., degree of supersaturation), and the process stability during scale-up are still unclear.

In general, crystallization is the final step of API manufacturing after a series of synthetic steps. It renders good control over crystal attributes of APIs such as polymorphism, purity, yield, and size distribution. Batch crystallization processes are widely used in the chemical and related industries, particularly the pharmaceutical industry, since they are versatile and relatively inexpensive without high capital investments. However, several challenges, including batch-to-batch variability and processing inefficiency, still remain in the pharmaceutical industry [57]. Contrasting with batch crystallization, continuous crystallization can provide high efficiency and reproducibility, good heat and mass transfer, smaller equipment size, and lower expenditure, waste generation, and variability during scale-up [58,59,60]. While the differences between batch and continuous processes exist in heat and mass transfer, attrition, and residence time distribution, the more desirable and consistent attributes may be achieved in continuous crystallization [61].

In a batch crystallizer, nucleation is generally carried out at a high degree of supersaturation first, and then the growth of crystals will dominate to consume the nutrient. The polymorphic transformation will happen when the solute concentration reaches the metastable zone of a stable form. As all crystallization events are happening in the same tank, once the stable polymorph nucleates later in time, the already-existing metastable form will be rapidly converted to the stable form by solution-mediated polymorphic transformation. Batch crystallization usually ends up at equilibrium, which gives sufficient time for polymorphic transformation to take place. For continuous crystallization, however, events can occur at non-equilibrium at any time. It is feasible to create a conducive crystallization condition for a metastable polymorph by changing the temperature or adding antisolvent at each segment of a plug flow reactor, or at each stage of a cascade of continuously stirred tanks. The crystals in metastable form can then be continuously isolated from solution before undergoing polymorphic transformation. Although the benefits of shifting from batch to continuous production in the future have been demonstrated by numerous examples [57], efforts and time are required for promoting continuous manufacturing in the pharmaceutical industry.

OXA, MAL, FUM, and benzoic acid are carboxylic acids and are commonly used as co-formers. To further investigate the relationship among additives and PCA polymorphs, several additives including the co-formers, which had been reported to form co-crystals with PCA, and some other ones which had not yet been found but belong to carboxylic acids, are chosen as the candidates to control the polymorphic formation of PCA in the present study. Firstly, PCA will be recrystallized by cooling crystallization in the presence of various additives to screen which additives could induce Form II PCA. Secondly, the influence of the amount of additives, the degree of supersaturation, and seeding on the polymorphic formation of PCA will be investigated. Lastly, the polymorphic crystallization with the assistance of additives will be carried out by batch crystallization in a stirred tank, and by continuous crystallization in a tubular crystallizer, to study the feasibility of scale-up. Figure 2 displays the molecular structures of PCA and all additives used in the present study.

## 2. Experimental Section

### 2.1. Materials

Adipic acid (ADI) (Lot MKBF9110V from USA), caffeine (CAF) (Lot 099k1441 from China), citric acid (CIT) (Lot 057K0090 from USA), *DL*-malic acid (MLC) (Lot MKBS7851V from China), sodium acetate (NaAc) (Lot SLBT2099 from Japan), succinic acid (SUC) (Lot 058K0706 from Switzerland), and theophylline (THP) (Lot MKCJ8498 from China) were purchased from Sigma-Aldrich. Fumaric acid (FUM) (Lot 10202035 from Canada), glutaric acid (GLU) (Lot 10186568 from UK), malonic acid (MAO) (Lot 10146415), and *DL*-tartaric acid (TAR) (Lot 10103590) were received from Alfa Aesar. Maleic acid (MAL) (Lot A0414668 from Spain) was obtained from Acros Organics. Sodium phosphate monobasic monohydrate (Lot E07165 from USA) was purchased from J.T. Baker. PCA (Lot 0540385 from China) was received from Anqiu Lu’an Pharmaceutical Co., Ltd. Phosphoric acid (aq) (86% assay, Lot. Ts10510B from Taiwan) was obtained from Echo Chemical Co., Ltd. Oxalic acid (OXA) dihydrate (Lot H4P111 from South Korea) was purchased from Daksun Pure Chemicals Co., Ltd. All chemicals have a purity of ≥99% and were used as received without further purification. Reverse osmosis (RO) water was treated through a water purification system (model Milli-RO plus) from Millipore (Billerica, MA, USA).

### 2.2. Additive Screening for the Preparation of Form II PCA

Twelve compounds, including ADI, CAF, CIT, FUM, GLU, MAL, MLC, MAO, OXA, SUC, TAR, and THP, were screened as additives for the preparation of Form II PCA (Figure 2). Two screening methods were carried out in which PCA crystals were produced by cooling crystallization from 60 to 25 °C.

For Screening Method 1, the initial PCA concentration of each aqueous solution was fixed according to the solubility of PCA in pure water (i.e., 47.45 ± 2.60 mg/mL at 60 °C) [43]. An amount of 100 mg of PCA and an equimolar amount of each additive except CIT were added to a 7 mL scintillation vial. Unlike the other cases using 1:1, the molar ratio of PCA to CIT was changed to 2:1 due to the stoichiometric ratio of the existing 2:1 PCA–CIT co-crystal [62]. An amount of 2.2 mL of water was introduced to dissolve PCA and the additives used at 60 °C. The aqueous solution was further heated to 65 °C to ensure no invisible particles or nuclei in the solution and then cooled to 25 °C by moving it to another water bath pre-set at that temperature to create a temperature gradient for PCA crystallization. If there was no crystal formation for one day, the solution was placed in a refrigerator at around 5 to 10 °C until PCA crystals appeared. The PCA crystals produced were filtered and dried at 40 °C overnight.

For Screening Method 2, PCA and each additive were added at the same molar ratio as the one for Screening Method 1. Since the aqueous solubility of PCA was varied in the presence of additives, water was titrated dropwise through a micropipette into each vial with intermittent shaking until all solids were just dissolved at 60 °C. Then, the same procedure as Screening Method 1 was followed.

### 2.3. Effect of Additive Amount on the Polymorphic Formation of PCA

Given the hints by the above “Additive Screening” experiments, ADI, FUM, MLC, OXA, and SUC were selected to study the effect of different amounts of additives (i.e., the weight ratios of PCA to additive) on the polymorphic formation of PCA. An amount of 100 mg of PCA was added to 1.7 mL of water to have a concentration of 58.8 mg/mL. Different weights of 25, 50, 75, and 100 mg of ADI, MLC, and SUC were added as an additive, while 10, 20, 30, and 50 mg of FUM were introduced. In addition, 41.7, 83.4, 125, and 166.9 mg of OXA dihydrate, standing for 30, 60, 90, and 120 mg of OXA, respectively, were added. They were heated to 75 °C to form a homogeneous solution first and then swiftly cooled to 10 °C. The produced crystals were filtered and dried at 40 °C overnight.

### 2.4. Effect of Seeding on the Polymorphic Formation of PCA

Forms I and II PCA and those additives (i.e., ADI, FUM, MLC, OXA, and SUC) were used as seeds to induce the polymorphic crystallization of PCA. Pure Form II PCA crystals were prepared by reaction coupling [43]. After cooling from 75 to 10 °C for 2 min, 2 mg of seed crystals was fed to each PCA–additive aqueous solution with intermittent shaking. Those solutions remained homogeneous without crystallizing prior to seeding. Upon the addition, it was maintained at 10 °C for 20 min. The produced crystals were filtered and dried at 40 °C overnight.

### 2.5. Liquid-Assisted Grinding

15 mg of purchased Form I PCA was mixed with an equimolar amount of ADI, FUM, MLC, and SUC in an agate mortar and ground together with a pestle for 1 min. An amount of 20 μL of methanol was dropped into the mixture. The slurry mixture was continuously ground for about 10 min until dry. The dried solids were characterized by Fourier-transform infrared spectroscopy (FTIR) for phase identification.

### 2.6. Effect of the Degree of Supersaturation on the Polymorphic Formation of PCA with the Assistance of FUM and OXA

The aqueous solutions of PCA–FUM and PCA-OXA were prepared at 75 °C with various PCA concentrations ranging from 56.8 to 15.5 mg/mL, which are equivalent to *S* = 5.5 to 1.5 according to our determined solubility of PCA in water at 10 °C (i.e., 10.3 mg/mL). The weight ratios of PCA to FUM and PCA to OXA varied from 1:0.2 to 1:0.5 and from 1:0.6 to 1:1.2, respectively. All PCA–FUM and PCA-OXA solutions were rapidly cooled from 75 to 10 °C. The produced crystals were filtered and dried at 40 °C overnight.

### 2.7. Solubility Measurement in the Aqueous Solutions of FUM and OXA

The solubility values of FUM, OXA, and Form I PCA in water and Form I PCA in the aqueous solutions with various concentrations of FUM and OXA were measured at 10 °C. Given amounts of FUM, OXA, and Form I PCA were suspended individually in 10 mL of water at 10 °C. Each suspension was placed in a plastic water basin, which was connected with a Six-Position Magnetic Stirrer system and stirred for 1 day to achieve equilibrium. On the other hand, aqueous stock solutions were prepared with various concentrations of FUM and OXA from 0.25 mg/mL to their particular solubilities at 10 °C. A given amount of purchased Form I PCA was suspended in 10 mL of the aqueous stock solutions of FUM and OXA at 10 °C and the suspensions were stirred for 1 day. All suspensions were then filtered by a 0.22 μm polyvinylidene difluoride (PVDF) syringe filter and analyzed by high-performance liquid chromatography (HPLC).

### 2.8. Cake Washing for the Removal of FUM

The filter cake was rinsed with NaAc (aq) to remove FUM solids from Form II PCA crystals, which were produced by PCA crystallization in the presence of FUM. The solubilities of FUM and Form I PCA in 0.5, 1, and 1.5 M NaAc (aq) were determined at 10 °C by the above-mentioned method. An amount of 1 g of PCA and 0.5 g of FUM were dissolved in 17 mL of water at 75 °C to prepare a PCA–FUM aqueous solution. The PCA–FUM solution was cooled to 10 °C at a cooling rate of 1 °C/min for 90 min. The produced solids were filtered by cake filtration using a ceramic Büchner funnel and a filter paper with a pore size of 5 μm. An amount of 15 mL of 1 M NaAc (aq) pre-chilled in a 10 °C water bath was poured into the funnel for cake washing. The liquid level should exceed the thickness of a filter cake. The wet cake was rinsed with the NaAc (aq) four times and the time for each rinsing was 5 s. A few milligrams of the wet cake were sampled for HPLC and powder X-ray diffraction (PXRD) analysis. For every washing (rinsing) with NaAc (aq), 15 mL of cold water was used to exchange the remaining NaAc (aq) on the wet cake for a solvent swap. Finally, the cake was dried in a 40 °C oven overnight.

### 2.9. Batch Additive-Assisted Cooling Crystallization of PCA in a Stirred Vessel

A total of twenty experiments listed in Appendix A were carried out in a 0.5 L jacketed glass vessel (Appendix A), which was equipped with an overhead agitator and a four-bladed (45° downflow) Teflon impeller. The clearance between the impeller and vessel bottom was set to the one-third liquid level. For each, 20 g of purchased PCA and 250 mL of water were added to the vessel under agitation at 300 rpm first. To dissolve all solids, the temperature was increased to 75 °C. An amount of 90 mL of water was then introduced to the solution to flush down the particles that remained on the wall or agitator. After complete dissolution, the homogeneous solution was cooled from 75 to 10 °C at a cooling rate of 1 °C/min. The operational variables, including agitation speed (300, 200, 100, and 0 rpm), the type of additive (FUM, OXA, or none), and the weight ratio of PCA to additive (1:0.2 and 1:0.5 for FUM and 1:0.6 and 1:1.2 for OXA) were tabularized in Appendix A. When PCA started to nucleate and form crystals, sampling was performed every five to ten minutes for examination by optical microscopy (OM).

### 2.10. Continuous Additive-Assisted Cooling Crystallization of PCA in a Tubular Crystallizer

A tubular crystallizer comprised nine jacketed glass tubes in the same length and inner diameter of 600 and 10 mm, respectively. Their inlet temperatures were recorded and labeled as T1 to T9 from top to bottom (Appendix A). Silicone tubing, which had a 20 cm length and a 9 mm inner diameter, served as a bridge between each tube. An aqueous solution of PCA was prepared by dissolving 20 g of purchased PCA in 340 mL of water in a stirred tank at 75 °C. The homogeneous solution was then fed into the tubular crystallizer by a peristaltic pump through a different silicone tube, which had a 100 cm length and a 7 mm inner diameter. A filtration apparatus was set up right below the outlet of the ninth (last) glass tube to collect crystals upon discharge. Some crystals that remained inside the tubular crystallizer were purged out with cold water and collected at the end. The operational variables, including flow rate (150 and 75 mL/min), the weight ratio of PCA to FUM (1:0.2, 1:0.3, and 1:0.5), and the initial concentration of PCA (58.8 and 44.1 mg/mL), are listed in Appendix A.

### 2.11. Instrumental Analysis

Optical Microscopy (OM). Two optical microscopes (Olympus SZII and BX-51, Tokyo, Japan) were used to observe crystal habits and forms.

Fourier-Transform Infrared Spectroscopy (FTIR). FTIR (Perkin Elmer Spectrum One, Norwalk, CT, USA) was used to identify organic compounds. Each solid sample was ground with KBr powder at a weight ratio of 2:98 to form a tablet using a hydraulic press, which was scanned in the region of 2000 to 400 cm^−1^ with a resolution of 2 cm^−1^ for 8 repetitions.

Powder X-ray Diffraction (PXRD). PXRD (Bruker D8 Advance, Karlsruhe, Germany) with an X-ray source of CuK*α* (*λ* = 1.542 Å) was used to determine crystallinity, polymorphism, and structural purity. The diffractometer was operated at 40 kV and 40 mA to generate diffraction patterns at a scanning rate of 0.05° 2*θ*/s from 2*θ* = 5 to 35°.

High-Performance Liquid Chromatography (HPLC). HPLC (Shimadzu Prominence-I LC-2030C 3D Plus, Japan) was used to measure the equilibrium concentrations (i.e., solubility) of PCA, FUM, and OXA. A Kinetex^®^ F5 column (150 mm × 4.6 mm × 4.6 μm particle size × 8.8 nm pore diameter) and an autosampler were installed in the HPLC system. A mobile phase of phosphoric acid buffer solution was pumped at a flow rate of 1 mL/min at 25 °C. The mobile phase was prepared by dissolving 40 mmol of sodium phosphate monobasic monohydrate and 10 mmol of 85% phosphoric acid in 800 mL of water. Water was then added to a total solution volume of 1 L at a pH of 2.6. The UV wavelength was set at *λ* = 210 nm.

^1^H and ^13^C Nuclear Magnetic Resonance Spectroscopy (NMR). NMR (Bruker Ascend 600 MHz, Germany) was used to identify molecular structures and define their relative stoichiometric ratio(s). An amount of 20 mg of each sample was dissolved in 1 mL of deuterated dimethyl sulfoxide (DMSO-d_6_).

## 3. Results and Discussion

### 3.1. Additive Screening for the Preparation of Form II PCA

Twelve compounds have been screened as additives for the preparation of metastable Form II PCA in the present study, including ADI, CAF, CIT, FUM, GLU, MAL, MAO, MLC, OXA, SUC, TAR, and THP, by Screening Methods 1 and 2 (Figure 2). Among those additive candidates, CAF, CIT, MAL, OXA, and THP could form co-crystals with PCA in a definite stoichiometric ratio [62,63,64,65,66]. For Screening Method 1, the initial concentration of PCA was identical in all cases regardless of the additive used. Since the solubility of PCA can be enhanced with the assistance of the additives, the initial concentration of PCA was varied and determined by water titration until all solids were just dissolved for Screening Method 2.

The results are summarized in Table 1 based on PXRD (Figure 3a). Noticeably, there was no crystal formation in the case of CAF for Screening Method 1 even though the solution was left in a refrigerator for several days. This indicated that the solubility of PCA was significantly increased and/or its metastable zone width (MZW) was much broadened in the presence of CAF. The PXRD patterns of the crystals harvested by Screening Methods 1 and 2 are displayed in Figure 3a,b, respectively. The corresponding FTIR spectra are also provided in Appendix A. When the solubility values of ADI and FUM in water or aqueous solution are close to or lower than that of PCA [67,68], the two additives crystallized out simultaneously with PCA in Table 1. Irrespective of the screening method used, only Form I PCA was obtained by the cooling crystallization of PCA in the presence of CIT, GLU, MAO, and TAR (Table 1). By varying the initial concentration of PCA in PCA–additive aqueous solutions for Screening Method 2, some cases gave different results from those for Screening Method 1. Form II PCA could be produced in the presence of FUM, MAL, MLC, OXA, and SUC as an additive for Screening Method 2, but only ADI, FUM, and SUC for Screening Method 1. As mentioned above, the cases of ADI and FUM resulted in the mixture of PCA and the respective additive (i.e., ADI or FUM). In the case of SUC, pure Form II PCA was obtained for both methods. In addition, new indications of a PCA–CAF co-crystal appeared in the FTIR spectra and PXRD patterns (Appendix A) for the case of CAF. They agreed with the results in the literature [69]. Unlike the cases of ADI and FUM, a mixture of PCA–THP co-crystal and Form II THP was obtained as a result because THP has a relatively lower solubility than PCA in an aqueous medium.

The aqueous solubility of PCA at 60 °C was increased to 77.5, 502.5, 55.3, 56.0, 54.8, 60.9, 52.3, and 169.2 mg/mL in the presence of ADI, CAF, GLU, MAL, MLC, OXA, SUC, and THP, respectively, as additives based on Screening Method 2, in comparison to 47.45 mg/mL in the absence of any additive. On the contrary, it was lowered to 27.8 mg/mL in the presence of FUM, probably due to the low aqueous solubility of FUM. There was no significant change in the aqueous solubility of PCA as determined to be 46.7, 47.8, and 44.3 mg/mL when CIT, MAO, and TAR, respectively, were present as an additive. It seems that PCA has a stronger interaction with CAF, OXA, and THP that can serve as a co-crystal former (i.e., co-former). Their presence or assistance contributes to the much-enhanced solubility of PCA and a solution complex may exist between them [70,71]. Apparently, the polymorphic formation of PCA is sensitive to the initial concentration (i.e., degree of supersaturation) and the additive type.

In general, crystal habit or morphology can be modified by the modulation of several factors, such as solvent, pH, seeding, polymorph, and the existence of impurities, additives, or foreign substances. Forms I and II PCA prefer to grow into tabular and needle shapes, respectively, in pure water. As shown in Figure 4a,h, tabular Form I and rod-like Form II PCA crystals were grown, respectively, while ADI crystals exhibit a plate-like or scalenohedral shape. FUM was crystallized as plate-like flakes and easily aggregated on the Form II PCA needle crystals in Figure 4b,j. It is noticeable that the crystals produced from the PCA-MAL solution by Screening Method 2 were yellow in color (Figure 4k), but those colored crystals were not observed for Screening Method 1 (Figure 4c) or in the other cases. Besides, the yellow needle-like crystals were radiated from small aggregates as circled in Figure 4k. A trace of FUM, presumably in the aggregates, was found due to the isomerization of MAL as evidenced by NMR (Appendix A). To prepare PCA-additive solutions with the initial PCA concentration close to its saturation point at 60 °C, water was added dropwise in ten-minute intervals for Screening Method 2. The relatively long experimental time at such a high temperature facilitated an unexpected reaction in the PCA–MAL aqueous solution.

The OM images in the cases of MLC (Figure 4d,l) and OXA (Figure 4e,m) also agree with the summarized results in Table 1 that are based on the PXRD patterns in Figure 3a. More uniform rod-shaped Form II PCA crystals were formed in the presence of SUC for Screening Methods 1 and 2 (Figure 4f,n). Both PCA–CAF and PCA–THF co-crystals have a needle-like shape (Figure 4g,i,o), as evidenced by their PXRD patterns in Appendix A, respectively. Besides, liquid-assisted grinding was applied to confirm the formation of PCA co-crystals due to its high efficiency in co-crystal screening [63]. There was no indication of co-crystal formation in all the cases of PCA–ADI, PCA–FUM, PCA–MLC, and PCA–SUC based on FTIR.

Performing the two screening methods on PCA crystallization in the presence of the five additives, ADI, FUM, MLC, OXA, and SUC, revealed the potential to control the selective formation of Form II PCA through the assistance of dicarboxylic acids. Despite the ambiguous mechanism, those additives possess a straight chain with an even number of carbon atoms and a dicarboxylic acid group (Figure 2). One tricarboxylic acid (i.e., CIT), and the other dicarboxylic acids having an odd number of carbon atoms (i.e., MAO and GLU) or a high steric effect thanks to hydroxyl groups on side chains (i.e., TAR), did not assist in the polymorphic formation of Form II PCA. To have a better understanding of the additive effect on the polymorphic formation of PCA, other factors such as the amount of additive, degree of supersaturation, and agitation rate will be explored in-depth in the later sections.

### 3.2. Effect of Additive Amount on the Polymorphic Formation of PCA

By employing a structurally related additive or a co-former as a guest to control the polymorphic form of a host molecule, the weight ratio of the host to the guest was considered. The weight ratio of PCA to additive showed a decisive influence on the polymorphic formation of PCA in Table 2 except in the case of MLC. The presence of MLC did not aid in the selective formation of Form II PCA at all, with the weight percentage ranging from 25 wt% to 100 wt% in this experiment. Form I PCA was favorably produced at a relatively low weight percentage of additive in the other cases.

In the cases of ADI and FUM, Form II PCA began to form at weight percentages of ≥50 and ≥20 wt%, respectively, as shown in Table 2. However, it was accompanied by ADI or FUM crystals due to their lower solubilities. When the weight percentage of ADI was increased to 75 and 100 wt%, Form I PCA was generated along with the mixture of Form II PCA and ADI crystals. We hypothesized that the precipitation of ADI led to a decline in the concentration of ADI and sequentially induced the nucleation of Form I PCA or polymorphic transformation from Form II to Form I.

There was a watershed between the weight percentages of 10 and 20 wt% of FUM for the selective formation of Form II PCA (Table 2). Form I PCA crystals were formed with 10 wt% of FUM and a mixture of Form II PCA and FUM could be harvested together without any trace of Form I PCA at 20 to 50 wt% of FUM. It was observed that Form II crystals were formed in the presence of FUM at 20 and 30 wt% before FUM started to precipitate. On the contrary, FUM crystals appeared first at 50 wt%, following the formation of Form II crystals. The results gave a mixture of Form II PCA and FUM above 20 wt%.

Since OXA is able to form a co-crystal with PCA and has a high aqueous solubility, the higher weight percentages of 30, 60, 90, and 120 wt% of OXA with respect to PCA, corresponding to the molar ratios of 1:0.5, 1:1, 1:1.5, and 1:2 of PCA to OXA, respectively, were chosen in the present study. Form II PCA was formed with part of Form I when the weight percentage of OXA was increased to 90 wt% (Table 2). Until it reached 120 wt%, pure Form II PCA was obtained instead. Following the same trend in the case of OXA, a mixture of Forms I and II PCA was received at 75 wt% of SUC, whereas pure Form II PCA survived at 100 wt% of SUC. In addition to the polymorphic selectivity, a delay in time for the birth of PCA crystals from a few minutes to several days would depend on the additive amount in the cases of OXA and SUC. The higher the weight percentage of the additive was, the longer the induction time of PCA was required.

No matter which type of additive was employed, the produced Forms I and II PCA crystals exhibited tabular and needle-like shapes, respectively (Appendix A). As evidenced, they were consistent with the results based on FTIR and PXRD. Additionally, metacetamol had been used as a tailor-made additive to control the formation of Form II PCA by altering the preferential growth of PCA polymorphs [72]. However, such a strategy failed to prepare Form II PCA in the present study.

### 3.3. Effect of Seeding on the Polymorphic Formation of PCA

According to the results of “Effect of Additive Amount”, the use of additives has a profound effect on the nucleation of PCA polymorphs. Furthermore, Forms I and II PCA and the five additives (i.e., ADI, FUM, MLC, OXA, and SUC) were utilized as a seed to induce the polymorphic crystallization of PCA in the present study. The induction time was significantly reduced from days to seconds by seeding in the cooling crystallization of PCA with and without the assistance of additives in all the cases regardless of the type of seed. It implied that seeding is a way to alter the crystallization kinetics. Their product compositions (without the assistance of additives) were revealed based on the PXRD patterns in Figure 5. It was not surprising that 100% Form I PCA was produced by seeding the aqueous solution of PCA with Form I PCA crystals (Figure 5a). Although seeding with Form II PCA crystals was capable of triggering the occurrence of nucleating Form II PCA, a trace amount of Form I PCA was found, as shown in Figure 5b. It was difficult to prepare pure Form II PCA crystals solely by seeding at a relatively high degree of supersaturation in aqueous solution. The polymorphic transformation from metastable Form II to stable Form I could be avoided by seeding at a relatively low temperature and concentration of PCA [32]. As a consequence, pure Form I PCA crystals were produced by adding all the additives as seeds (Figure 5c,e–g), except FUM (Figure 5d). As for seeding with FUM, the characteristic diffraction peaks for Form I at 2*θ* = 12.2, 13.8, and 15.6° and Form II at 2*θ* = 10.3° were detected in the product crystals, as shown in Figure 5d.

Concerning seeding in the PCA–additive aqueous solutions (i.e., with the assistance of additives) with the corresponding additives as a seed, a mixture of Forms I and II PCA was formed in the case of ADI (Figure 6a), in agreement with to the outcome of “Effect of Additive Amount” at 100 wt% of ADI (Table 2). The presence of ADI crystals could induce the nucleation of Form II PCA. However, its polymorphic transformation to Form I was followed in the case of ADI, where a trace amount of Form I PCA was noticed, as shown in Figure 6a. The formation of Form I PCA crystals might also be attributed to the inevitable polymorphic transformation happening in the case of seeding in the PCA–FUM solution with FUM (Figure 6b). Seeding in the PCA–MLC and PCA-OXA solutions, respectively, with MLC and OXA, brought about the formation of Form I PCA alone (Figure 6c,d). Remarkably, pure Form II PCA crystals were received by seeding with SUC, as evidenced in Figure 6e.

### 3.4. Effect of the Degree of Supersaturation on the Polymorphic Formation of PCA with the Assistance of FUM and OXA

Among the five additives, FUM and OXA were selected to study the effect of the degree of supersaturation on the polymorphic formation of PCA since FUM is the most stable additive for making Form II PCA, and OXA could serve as a co-former for preparing a PCA co-crystal (Table 1). SUC was excluded for the reason that its use failed to prepare Form II PCA upon scaling up to a gram scale. The degree of supersaturation, *S*, is expressed by *C*/*C**, where *C* is the concentration of a solute and *C** is the equilibrium concentration of the solute (i.e., solubility) under certain conditions. A dependence of *S* on the polymorphic form of PCA in water has been reported [29]. PCA prefers nucleating as Form I at a relatively low *S* in water and following the order of Forms, II, III, and I as *S* increases.

In our study, different *S* were attempted for the additive-assisted cooling crystallization of PCA in the cases of FUM and OXA. Figure 7 indicates that the polymorphic outcome was dependent on *S* as well as the amount of additive. In the case of FUM, Form II PCA could be obtained at *S* = 1.5 to 3.0 at 20 wt% of FUM. When the initial concentration of PCA gave *S* = 1.5 without the assistance of an additive, the route for PCA crystallization would enter into the MZW of Form I, which overlaps with the solubility curve of Form II according to the solubility diagram described by Sudha and Srinivasan [28,29].

It was proposed that the presence of FUM could modulate the shape and width of an MZW and further facilitate the nucleation of Form II PCA (Figure 7). As *S* was below 3 in the case of FUM at 20 wt%, the concentration could enter into the metastable zones of Forms I and II PCA as well. However, the PCA–FUM system preferred producing Form II PCA in Figure 7. At *S* = 3 to 5.5, it would reach the labile zones of Forms I and II PCA. At a high *S*, the influences on the rates of nucleation and solution-mediated polymorphic transformation would become apparent. Form I PCA can still nucleate and grow to a visible size at a slow cooling rate, especially when the fast rate of cooling is hardly achieved on a large or industrial scale. For the case of FUM at 50 wt%, pure Form II PCA was produced in a region of *S* = 1.5 to 5.5 and the *S* region could be extended to 5.7 if the results in Table 2 are considered as well. At the lower 20 wt%, Form I PCA appeared at an *S* of ≥3.0 to form a mixture with Form II. An increase in the amount of FUM up to 50 wt% can stabilize the formation of Form II PCA over the wide *S* region.

With the assistance of OXA at 60 wt%, the nucleation of PCA gave a similar trend as that in the absence of additive [28,29]. PCA was crystallized as from Form I to a mixture of Forms I + II and then to pure Form II PCA when *S* increased in the absence of additive (Figure 7). According to Figure 7, when *S* was located in the range between 2.2 and 4.8, the stability for yielding Form II PCA was increased. Nevertheless, Form I PCA tended to form as *S* was further increased above 4.8. As for the case of adding 120 wt% of OXA (i.e., PCA:OXA = 1:2 mol/mol), the *S* ranges for Form I and the mixture of Forms I and II PCA were wider than the ones for 60 wt% of OXA (i.e., PCA:OXA = 1:1 mol/mol). Therefore, the effect of *S* was dependent on the type and the amount of additives contained in the solution.

In our study, the region for Form III PCA crystals was not observed due to its extremely fast transformation. However, with a plentiful amount of FUM or OXA, Form II PCA crystals could be prepared in a wider region of *S.* Our methods provided more tolerance to the concentration variance than without the assistance of additives. Moreover, the initial concentration of PCA could be substantially increased for a higher yield. In general, the *S* value gradually decreases with the generation of PCA crystals, and Form I PCA crystals were formed when *S* had reached the red range below 3 as illustrated by the top red bar in Figure 7. This concern can be alleviated by adding FUM because of a wider *S* range for the preferential crystallization of Form II PCA. As a consequence, our robust process could offer a higher capacity and production efficiency.

### 3.5. Solubility Diagrams of the PCA–FUM and PCA-OXA Aqueous Solutions

Figure 8 displays the solubility diagrams of PCA in the presence of FUM and OXA. As shown in Figure 8a, the solubility of PCA could be enhanced by a narrow margin of 10% as the concentration of FUM was increased. The linear correlation at the whole [FUM] region indicated no formation of a PCA–FUM co-crystal.

Since PCA can form a 1:1 co-crystal with OXA, the dissociation behavior of the 1:1 PCA-OXA co-crystal can be expressed as follows if a solution complex is not considered to exist [70,73,74]:(1)$${\mathrm{PCA}\text{-}\mathrm{OXA}}_{\left(s\right)}\;\overset{K_{\mathrm{sp}}}{↔}{\;\mathrm{PCA}}_{\left(\mathrm{aq}\right)}{+\mathrm{OXA}}_{\left(\mathrm{aq}\right)}$$
where *K*_sp_ is defined as:(2)$$K_{\mathrm{sp}}=\left[\mathrm{PCA}\right]\left[\mathrm{OXA}\right]$$

The solubility product of the co-crystal, *K*_sp_, can be estimated with a slope by plotting [PCA] vs. 1/[OXA] according to Equation (3):(3)$$\left[\mathrm{PCA}\right]=\frac{K_{\mathrm{sp}}}{\left[\mathrm{OXA}\right]}$$

A linear relationship is expected without a term for intercept when no solution complex is formed. If a solution complex is considered present in the solution, the equilibrium solubility of the co-crystal can further be expressed by [70,73,74]:(4)$${\mathrm{PCA}\text{-}\mathrm{OXA}}_{\left(s\right)}\;\overset{K_{\mathrm{sp}}}{↔}{\;\mathrm{PCA}}_{(\mathrm{aq})\;}{+\mathrm{OXA}}_{\left(\mathrm{aq}\right)}$$
(5)$${\mathrm{PCA}}_{\left(\mathrm{aq}\right)}{+\mathrm{OXA}}_{\left(\mathrm{aq}\right)}\;\overset{K_{11}}{↔}{\;\mathrm{PCA}\text{-}\mathrm{OXA}}_{\left(\mathrm{aq}\right)}$$
where *K*_11_ is a first-order complexation constant. *K*_11_ can be written as:(6)$$K_{11\;}=\frac{\left[\mathrm{PCA}\text{-}\mathrm{OXA}\right]}{\left[\mathrm{PCA}\right]\left[\mathrm{OXA}\right]}=\frac{\left[\mathrm{PCA}\text{-}\mathrm{OXA}\right]}{K_{\mathrm{sp}}}$$

When PCA and OXA in solution exist in two forms, individual (free) molecules and a PCA-OXA complex, their total concentrations are represented as:(7)$${\left[\mathrm{PCA}\right]}_{\mathrm{total}}=\left[\mathrm{PCA}\right]+[\mathrm{PCA}\text{-}\mathrm{OXA}]$$
(8)$${\left[\mathrm{OXA}\right]}_{\mathrm{total}}=\left[\mathrm{OXA}\right]+[\mathrm{PCA}\text{-}\mathrm{OXA}]$$

By combining Equations (2), (3) and (6)–(8), the *K*_sp_ of the co-crystal is presented as:(9)$${\left[\mathrm{PCA}\right]}_{\mathrm{total}}=\frac{K_{\mathrm{sp}}}{{\left[\mathrm{OXA}\right]}_{\mathrm{total}}-{\;K}_{11}K_{\mathrm{sp}}\;}{+K}_{11}K_{\mathrm{sp}}$$
when [OXA]_total_ >> *K*_11_*K*_sp_, Equation (9) can be rewritten to become:(10)$${\left[\mathrm{PCA}\right]}_{\mathrm{total}}=\frac{K_{\mathrm{sp}}}{{\left[\mathrm{OXA}\right]}_{\mathrm{total}}}{+K}_{11}K_{\mathrm{sp}}$$

This implies that the existence of a solution complex will affect the solubility behavior. The dependence of the co-crystal solubility on the concentrations of PCA and OXA is shown in Figure 8b. The solubility data in the range of 0.01 M to 0.4 M of [OXA]_total_ were fitted based on Equation (10), and therefore, the *K*_sp_ and *K*_11_ of PCA-OXA co-crystals were estimated using the slope and intercept of the fitting equation in Figure 9 to be 4.15 × 10^−4^ M^2^ and 153.91 M^−1^, respectively, with an R square of 0.83. The large intercept indicates the formation of a solution complex. However, the solubility points at higher concentrations of OXA shifted away from the fitting curve (Figure 8b).

When [OXA]_total_ was increased above 0.5 M, the slope (i.e., Δ[PCA]_total_/Δ[OXA]_total_) became positive, leading to a concave upward (U-shaped) solubility curve in Figure 8b. Unlike the present study (all experiments were conducted in water), the *K*_sp_ of PCA-OXA co-crystals in acetonitrile was decreased with an increase in [OXA]_total_, until [OXA]_total_ had reached the solubility of OXA [66]. The upward solubility curve is associated with the formation of a high-order solution complex. Nehm et al. had proposed various solubility models for multiple complexes existing in the liquid and solid phases [70]. The solubility trend contributed by multiple complexes would be different from that of a single complex. PCA-OXA_2_ and PCA_2_-OXA were the two possible complexes if a second-order solution complex was formed. However, the trend of Form II PCA usually crystallized in a high concentration of OXA implied that the new solution complex associated with Form II PCA formation should appear in a similar environment. Therefore, in our study, a PCA-OXA_2_ solution complex was considered rather than a PCA_2_-OXA complex. A PCA_2_-OXA solution complex might appear in the left region of a 1:1 solution complex (if it existed), but it was less related to Form II PCA. If a 1:2 complex (i.e., PCA-OXA_2_) exists, a different complexation constant, *K*_12_, would be applied. [PCA]_total_ is then given as:(11)$${\left[\mathrm{PCA}\right]}_{\mathrm{total}}=\left[\mathrm{PCA}\right]+\left[\mathrm{PCA}\text{-}\mathrm{OXA}\right]{+[\mathrm{PCA}\text{-}\mathrm{OXA}}_{2}]$$
(12)$${\left[\mathrm{PCA}\right]}_{\mathrm{total}\;}=\frac{K_{\mathrm{sp}}}{{\left[\mathrm{OXA}\right]}_{\mathrm{total}}}{+K}_{11}K_{\mathrm{sp}\;}{+K}_{11}K_{12}K_{\mathrm{sp}}{\left[\mathrm{OXA}\right]}_{\mathrm{total}}$$

Accordingly, the plot of [PCA]_total_ vs. [OXA]_total_ is a concave upward curve (U-shaped) and the dependence of co-crystal solubility on their concentration will be significant at high [OXA]_total_. The existence of a high-order solution complex is suggested in Figure 8b. In addition, the increase in the solubility of PCA was responsible for the long induction time, especially at high concentrations of OXA.

Form II PCA preferred to form at 120 wt% than 60 wt% of OXA. The similarity in crystal structure between Form II PCA and 1:1 PCA-OXA co-crystal suggests the selective polymorphic formation of PCA in the OXA-assisted aqueous solution [56]. In this regard, the high-order PCA-OXA solution complex might possess a resembling molecular arrangement in water and thus favor the generation of Form II PCA nuclei over a wide *S* range. The PCA-OXA solution complexes could serve as a template to lead the PCA-free molecules to arrange toward Form II. A similar solution complex between PCA and FUM was not observed in the present study. It was speculated that the FUM molecule could facilitate and inhibit the Form I and Form II PCA nucleation. However, the interaction might not be identical to the co-crystal. Based on the results above, the carboxylic acid group and the stereoscopic structure seemed to be the keys to the selectivity of polymorphic formation.

### 3.6. Removal of FUM Crystals from the Mixture of Form II PCA and FUM by Solvent Rinsing

Although seeding with FUM seems to be the most reliable selection for Form II PCA, the formation of Form II PCA crystals was always accompanied by the precipitation of FUM. Based on our experimental findings, the stability of Form II PCA was related to the weight percent of FUM (Table 2). More than 20 wt% of FUM as an additive would be needed to prepare metastable Form II PCA reproducibly. As a consequence, the precipitation of FUM was bound to occur simultaneously.

Solvent rinsing is required to remove impurities from a filter cake after the crystallization of the desired product. In general, several requirements for a proper rinsing solvent include: (1) miscible with mother liquor, (2) insoluble to the desired product but highly soluble to impurities, and (3) without inducing subsequent, uncontrollable nucleation during rinsing [75]. None of the common solvents could meet the requirements based on our preliminary results. It was reported that the aqueous solution of NaAc could be used to inhibit the polymorphic formation of PCA because of the reduced solubility [43]. Moreover, FUM shows good solubility in NaAc (aq). Therefore, the NaAc (aq) was employed as a rinsing solvent to remove FUM existing in Form II PCA crystals upon filtration.

The solubility of FUM was first determined at 10 °C, as shown in Figure 10a. It was greatly enhanced from 3.38 ± 0.25 mg/mL in pure water to 55.53 ± 2.56 mg/mL in 1 M NaAc (aq) (Figure 10a). In contrast to FUM, the solubility of Form I PCA decreased from 10.32 ± 0.31 mg/mL in water to 7.89 ± 0.64 mg/mL in 1 M NaAc (aq). The maximum difference or disparity in solubility between PCA and FUM exists when 1 M NaAc (aq) is used in Figure 10a. The larger the solubility difference between FUM and PCA, the more conducive it was to the purification of Form II PCA. As a result, 1 M NaAc (aq) was chosen as a rinsing solvent.

When PCA was crystallized from the PCA–FUM aqueous solution at 50 wt% of FUM (Table 2), a mixture of Form II PCA and FUM was obtained with yields of 64.17 ± 2.25 and 71.15 ± 2.29%, respectively. The product composition on a filter cake after rinsing with 1 M NaAc (aq) is shown in Figure 10b. As the number of rinses increased, the content of FUM in the product was significantly reduced. Almost no trace of FUM was detected after rinsing thrice (Figure 10b). In addition to chemical purity, polymorphic purity was also examined by PXRD. The PXRD patterns in Figure 11 confirmed that FUM was thoroughly removed after three or four rinses without the polymorphic transformation from Form II to Form I PCA. This implies that FUM was not incorporated into the crystal lattice of PCA. As reported, metacetamol has an excellent performance in controlling the polymorphic formation of Form II PCA because it can be incorporated into PCA and thus selectively restrain the nucleation and growth of Form I PCA [44,54]. However, incorporation of impurities or foreign species into the crystal lattice of a given product can make impurity removal by common purification strategies, such as recrystallization and solvent rinsing, very difficult under the premise of ensuring the structural purity of Form II PCA. Therefore, the removal of FUM from the Form II wet filter cake evidenced that the incorporation of FUM molecules in the PCA crystal lattice did not occur at all, which has become the advantage of using FUM as an additive.

### 3.7. Batch Additive-Assisted Cooling Crystallization of PCA in a Stirred Vessel

Table 3 summarizes the results collected by the batch crystallization of PCA without the assistance of FUM or OXA in a 0.5 L stirred vessel. Since MZW is proportional to induction time [76], the MZW for the PCA–FUM and PCA-OXA cases would be compared in terms of induction temperature. When PCA crystals of a visible size started to appear at a higher temperature (i.e., induction temperature), it indicated a narrower MZW, as well as a shorter induction time. The MZW became narrower as the agitation speed was increased, as shown in Table 3. The higher agitation speed applied could also push the systems toward achieving equilibrium faster. Besides, the prolonged induction time was somehow attributed to an enhancement in the PCA solubility with the assistance of the additive. Since FUM precipitated earlier than PCA at 50 wt% of FUM, the induction temperatures in Expt. 7 to 9 were higher than those in Expt. 4 to 6 at 20 wt% FUM on average (Table 3). On the other hand, the presence of OXA in the PCA solution resulted in lower induction temperatures as compared to those with FUM and without any additive.

The OM images of the produced crystals right at the induction temperature, 5 min after induction, and at the end are shown in Appendix A for Expt. 1 to 3, 4 to 9, and 10 to 15, respectively. As anticipated, only Form I PCA crystals were harvested for Expt. 1 to 3 without the assistance of additive, although needle crystals of Form II PCA appeared at the beginning of crystallization at the agitation rates of 200 and 100 rpm (i.e., Expt. 2 and 3 in Appendix A). The metastable Form II PCA crystals were transformed into stable Form I crystals in 5 min.

Through the help of FUM at 20 wt%, needles of Form II PCA were seen around the induction time point at 200 and 300 rpm, while Form I PCA tabular crystals were seen at 100 rpm. It was relatively easy to prepare and stabilize Form II PCA in the presence of FUM above 20 wt% on a small scale (Table 2), but difficult to deliver the same results on a much larger scale of 0.5 L (Expt. 4–6, Table 3). A challenge was raised to control the crystallization of a metastable polymorph upon scaling-up. In addition, FUM did not precipitate until cooling to 10 °C for 60 min. When the weight percent of FUM was increased from 20 to 50 wt%, Form II PCA and FUM were simultaneously produced in needle- and flake-like shapes, respectively (Expt. 7–9, Appendix A), and existed for a longer time, except in Expt. 7. Once more FUM crystals were present in the environment, it became more favorable for Form II PCA to nucleate. However, the nucleation of Form I might dominate at a relatively low *S* when the concentration drop was consumed for the formation of Form II. The highest agitation speed of 300 rpm facilitated the polymorphic transformation from Form II to Form I PCA earlier in Expt. 7 (Appendix A) due to a faster mass transfer. All Form II PCA crystals were transformed into more stable Form I even though there was merely a trace of Form I crystals. Eventually, Form I PCA survived along with FUM crystals in all batches of Expt. 4–9 (Table 3).

The use of OXA as an additive showed a weaker effect on the selective formation of Form II PCA than that of FUM. The shift in induction time toward a lower temperature indicated that the existence of the PCA-OXA solution complex did affect the crystallization behavior of PCA, but did not help the nucleation of Form II much or not for too long on a larger scale under highly intensive agitation, regardless of the amount of OXA used in Expt. 10–15 (Table 3 and Appendix A).

The agitation had a strong impact on both crystallization kinetics and phase transformation kinetics. To slow down the rate of polymorphic transformation, Expt. 16 to 20 were designed to conduct the additive-assisted cooling crystallization of PCA without agitation in a batch mode (Table 3). Their induction temperatures were remarkably much lower than those with agitation applied (Table 3). The slow crystallization kinetics might lead to lower yields of PCA since a prolonged time was required to achieve equilibrium concentration for a good yield. Noticeably, Form II PCA remained in the FUM-containing solution for 4 h, as shown in Figure 12 (i.e., Expt. 17 and 18). When the crystallization of PCA occurred at a low temperature, the polymorphic transformation from Form II to Form I could also be retarded in Expt. 17 and 18.

### 3.8. Continuous Additive-Assisted Cooling Crystallization of PCA in a Tubular Crystallizer

Although Form II PCA could be formed with the assistance of FUM, its problematic polymorphic transformation remained an issue, especially when agitation is always required for better control in a large-scale stirred vessel. High shear and fast mass transfer might interfere with the molecular interactions between PCA and FUM. Therefore, continuous crystallization was investigated for the preparation of Form II PCA with the assistance of FUM using a tubular crystallizer. It comprised nine glass tubes. The solution temperature was thoroughly recorded during continuous crystallization. According to a temperature profile in the tubular crystallizer (Appendix A), the solution of PCA became supersaturated when it flowed into the third tube. PCA crystallization took place from the fourth to the ninth tubes. Rapid cooling caused the broadened MZWs of PCA and FUM. Since the residence time was limited and much shorter and PCA crystallized at a lower temperature in a continuous mode as compared to a batch mode, most of the solute molecules of PCA still remained in the solution, resulting in a low yield of PCA in Expt. 21–28 (Table 4).

The composition of the products collected directly from the outlet of the tubular crystallizer was examined by FTIR, PXRD, and HPLC. Moreover, the other products remaining inside the tubes were analyzed after discharge for comparison. In Expt. 21 to 24, the products were a mixture of Forms I and II PCA and FUM at 20 to 30 wt% of FUM. The amount of Form II crystals was proportional to the weight percent of FUM added.

In Expt. 25 at 50 wt% of FUM, the tubular crystallizer was clogged up with the massive PCA and FUM crystals precipitated, and thus no product could be discharged and collected from the outlet at a lower flow rate of 75 mL/min (Table 4). As the flow rate was raised to 150 mL/min, a mixture of Form II PCA and FUM crystals was obtained in Expt. 26. Based on Expt. 21 to 26, Form II PCA preferred forming at the higher flow rate of 150 mL/min (Table 4). A possible explanation is that the short residence time was unable to render the complete polymorphic transformation from Form II to Form I PCA. As mentioned above, the crystallization mainly occurred in the latter tubes of the tubular crystallizer. A very high *S* was created and caused the instant precipitation of PCA and FUM fine crystals, clogging up those tubes. Subsequently, their rapid growth into large crystals then exacerbated the clogging and solid settling. As a consequence, the yields of Expt. 21 to 26 were less than 1.4%. Therefore, the initial concentrations of PCA and FUM were lowered by a quarter in Expt. 27 and 28 to alleviate the clogging of precipitates by creating a smaller *S* and enabling slower rates of nucleation and crystal growth. It turned out higher yields of PCA of >3.1% (Table 4). The same trend was given that purer Form II PCA crystals were harvested at the higher flow rate of 150 mL/min along with FUM crystals.

In our study, a continuous crystallization process was successfully demonstrated for making metastable Form II PCA. The low yields and clogging indicated that most of the dissolved and crystallized PCA remained in the discharged fluid and tubular crystallizer. The crystals that remained in the tubes were comprised of Form II PCA and FUM in Expt. 26 and 28. If the tubular crystallizer and operating conditions could be properly redesigned to collect the remaining crystals in the tubes, the yield of Form II PCA would be enhanced. Apparently, operating conditions in a tubular crystallizer, such as reducing the initial concentration, increasing the amount of FUM addition, and flow rate, were conducive to the production of Form II PCA crystals.

## 4. Conclusions

A multicomponent crystallization method was developed to selectively produce Form II PCA by cooling crystallization with the assistance of additives in aqueous media. Given the additive screening results, Form II PCA could be yielded with ADI, FUM, MLC, OXA, and SUC. All additives gave a general trend that the larger the amount of additive added, the higher the probability of Form II PCA crystals being produced. Form II PCA crystals were harvested in the presence of 25 wt% of ADI, 20 to 50 wt% of FUM, 120 wt% of OXA, and 100 wt% of SUC. In the seeding experiments, only seeding with Form II PCA and FUM in the saturated PCA aqueous solution was capable of inducing Form II PCA crystals, while the other four additives had failed to do so. The *S* value for forming Form II PCA crystals was broader in the presence of OXA and FUM. The range could be altered from about *S =* 3.3 to 3.6 in the absence of additives and from *S* = 1.5 to 5.7 by using 50 wt% of FUM with respect to PCA. Different additives had different degrees of enhancement on the crystallization kinetics of Form I and Form II PCA. The solubility diagrams for PCA–FUM and PCA-OXA systems were constructed. In addition, a new high-order solution complex comprised of PCA and OXA or even water molecules, but different from the 1:1 complex of PCA-OXA, was implied by the concave upward solubility curve. Considering the similar molecular packings between Form II PCA crystals and PCA-OXA co-crystals, the new PCA-OXA solution complex might be closer to the arrangement of Form II PCA and tended to the preferential nucleation of Form II PCA crystals in aqueous solution. Although the complex was not seen in the PCA–FUM system, FUM, OXA, ADI, MLC, and SUC, having straight-chain dicarboxylic acid additives with an even number of carbons in common, exhibited a stronger effect on controlling the formation of PCA polymorphs. The special molecular structures might guide the PCA molecules to align to the arrangement of Form II nuclei in the aqueous solution.

A stirred tank for batch crystallization, and a tubular crystallizer for continuous crystallization, were used to prepare Form II PCA crystals by our method. The induction time was shortened as the agitation speed was increased; however, the polymorphic transformation was accelerated at the same time due to the high shear rate and mass transfer. On the other hand, continuous crystallization could be used to isolate the Form II PCA crystals by adjusting the flow rate and the initial concentrations of PCA and FUM. Form II PCA crystals were obtained at the initial concentrations of 20 g of PCA/340 mL of water and 15 g of PCA/340 mL of water with 50 wt% of FUM and a flow rate of 150 mL/min. Increasing the flow rate and the amount of FUM could enhance the probability of producing Form II PCA. The two challenges of crystal settling and low crystal yield would need to be overcome in the future.

The residual FUM in the products could be removed by solvent rinsing. The advantage of a remarkable solubility disparity between PCA and FUM in the NaAc aqueous solution was taken to dissolve the remaining FUM in the wet filter cake without worrying about the occurrence of the solution-mediated polymorphic transformation of Form II to Form I PCA. As compared to the method of adding metacetamol, which suffered from the incorporation of impurities in the crystal lattice, the additives used here, such as OXA, SUC, and FUM, could be easily separated from the final product. Screening the additives by starting from the co-formers was indeed feasible to select a proper additive for controlling metastable polymorphs. The application of a co-former is more advantageous than the structure-related additive for the purification of the desired product.

## Figures and Tables

**Figure 1 pharmaceutics-14-01099-f001:**
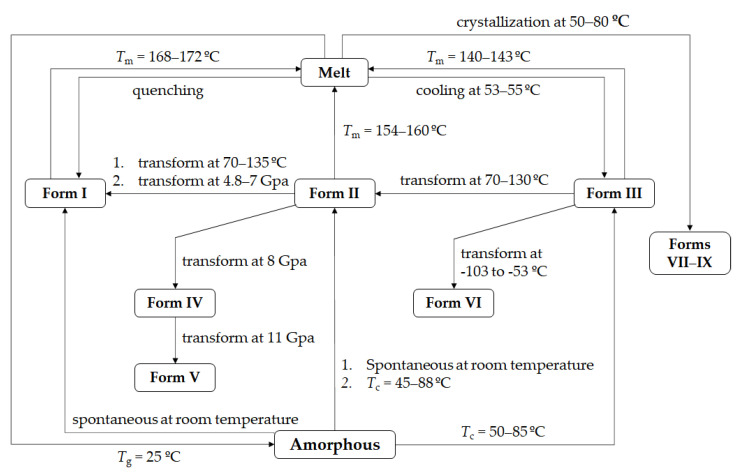
Phase transitions of nine PCA polymorphs (*T*_m_, melting temperature; *T*_c_, crystallization temperature; and *T*_g_, glass transition temperature).

**Figure 2 pharmaceutics-14-01099-f002:**
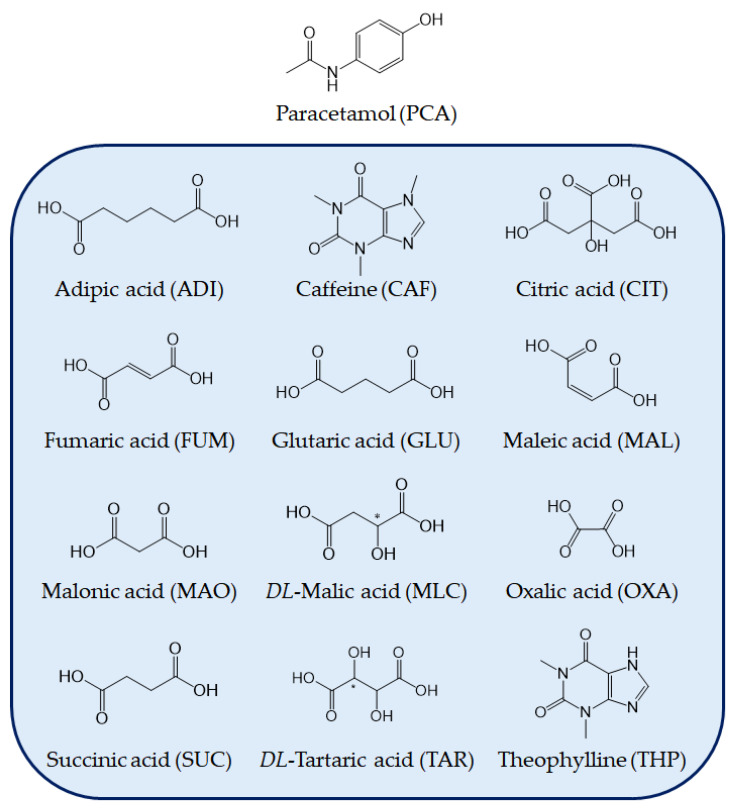
Molecular structures of PCA and the additives used in the present study.

**Figure 3 pharmaceutics-14-01099-f003:**
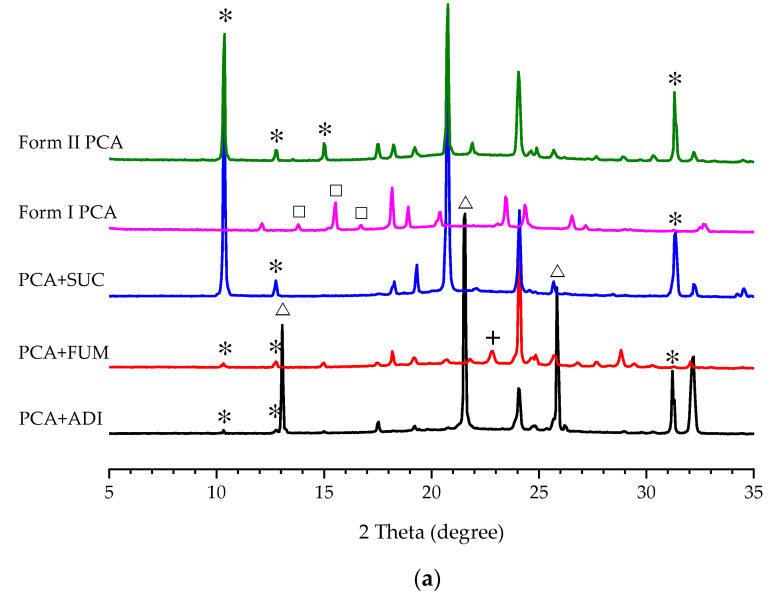
PXRD patterns of the PCA crystals produced by (**a**) Screening Method 1 and (**b**) Screening Method 2 with different additives as compared to Form I PCA (purchased) and Form II PCA (prepared by reaction coupling) on the top. The characteristic diffraction peaks of Form I PCA, Form II PCA, ADI, and FUM are labeled by ☐, ✴, △, and +, respectively.

**Figure 4 pharmaceutics-14-01099-f004:**
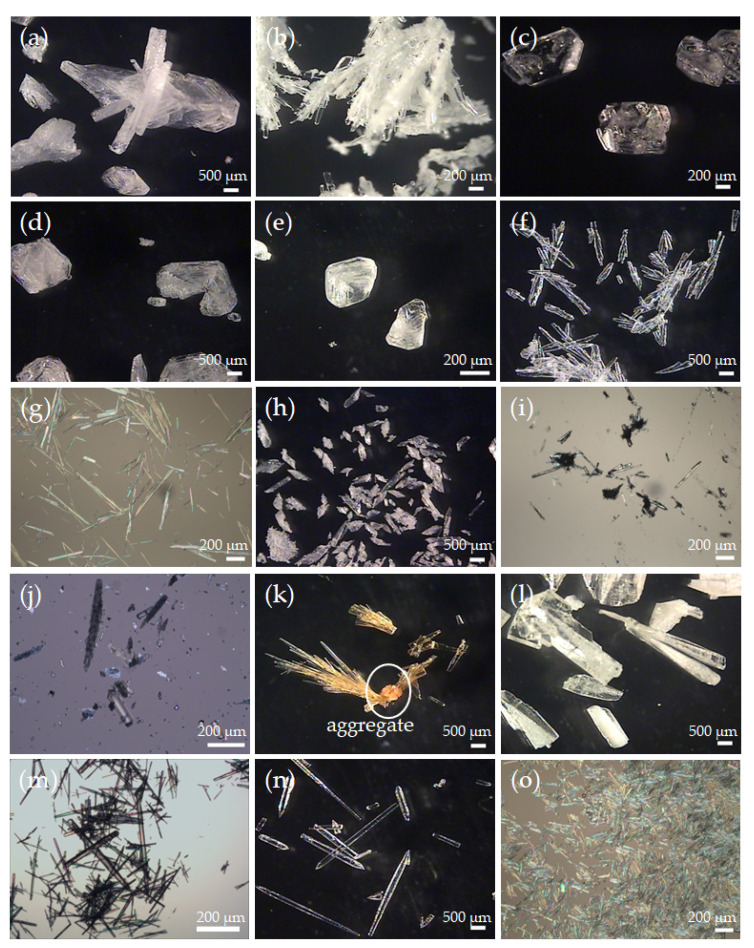
OM images of the PCA crystals produced by Screening Method 1 with (**a**) ADI, (**b**) FUM, (**c**) MAL, (**d**) MLC, (e) OXA, (**f**) SUC, and (**g**) THP, and by Screening Method 2 with (**h**) ADI, (**i**) CAF, (**j**) FUM, (**k**) MAL, (**l**) MLC, (**m**) OXA, (**n**) SUC, and (**o**) THP. The impurity in (**k**) is indicated by a circle.

**Figure 5 pharmaceutics-14-01099-f005:**
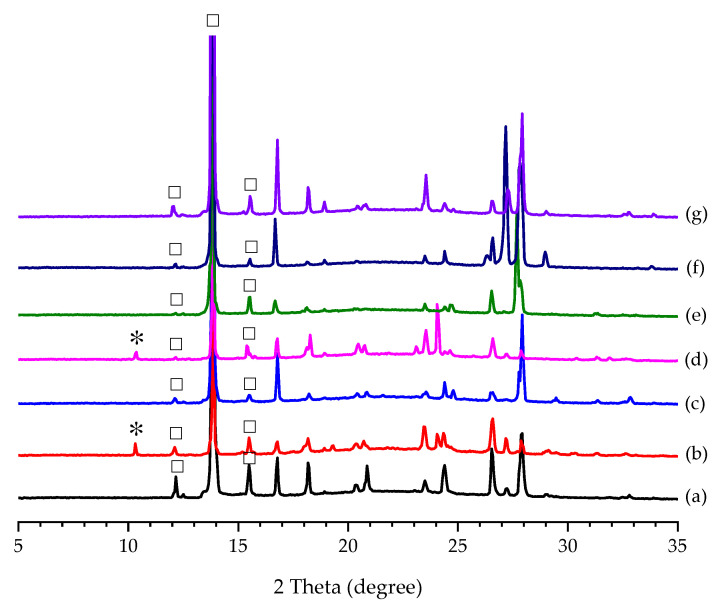
PXRD patterns of the PCA crystals harvested by seeding the PCA aqueous solutions with (**a**) Form I PCA, (**b**) Form II PCA, (**c**) ADI, (**d**) FUM, (**e**) MLC, (**f**) OXA, and (**g**) SUC. The characteristic peaks of Form I PCA and Form II PCA are labeled by ☐ and ✴, respectively.

**Figure 6 pharmaceutics-14-01099-f006:**
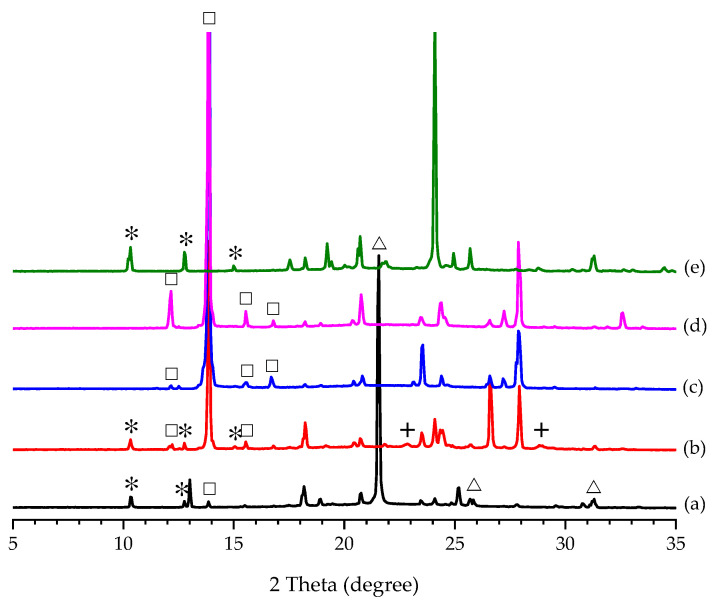
PXRD patterns of the PCA crystals harvested by seeding the PCA–additive aqueous solutions with (**a**) ADI, (**b**) FUM, (**c**) MLC, (**d**) OXA, and (**e**) SUC. The characteristic peaks of Form I PCA, Form II PCA, ADI, and FUM are labeled by ☐, ✴, △, and +, respectively.

**Figure 7 pharmaceutics-14-01099-f007:**
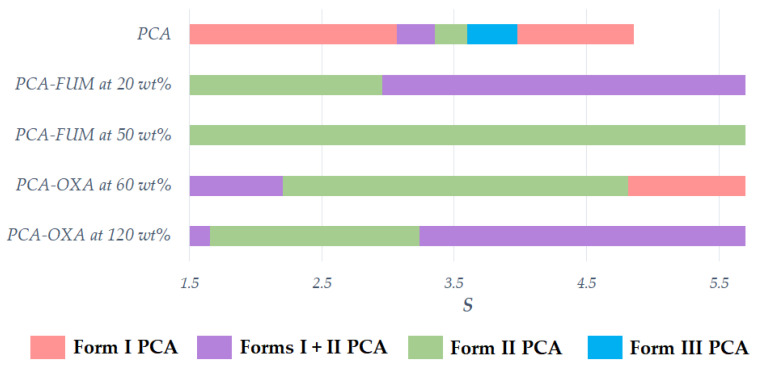
The types of PCA polymorphs with respect to the degree of supersaturation of PCA in aqueous solutions.

**Figure 8 pharmaceutics-14-01099-f008:**
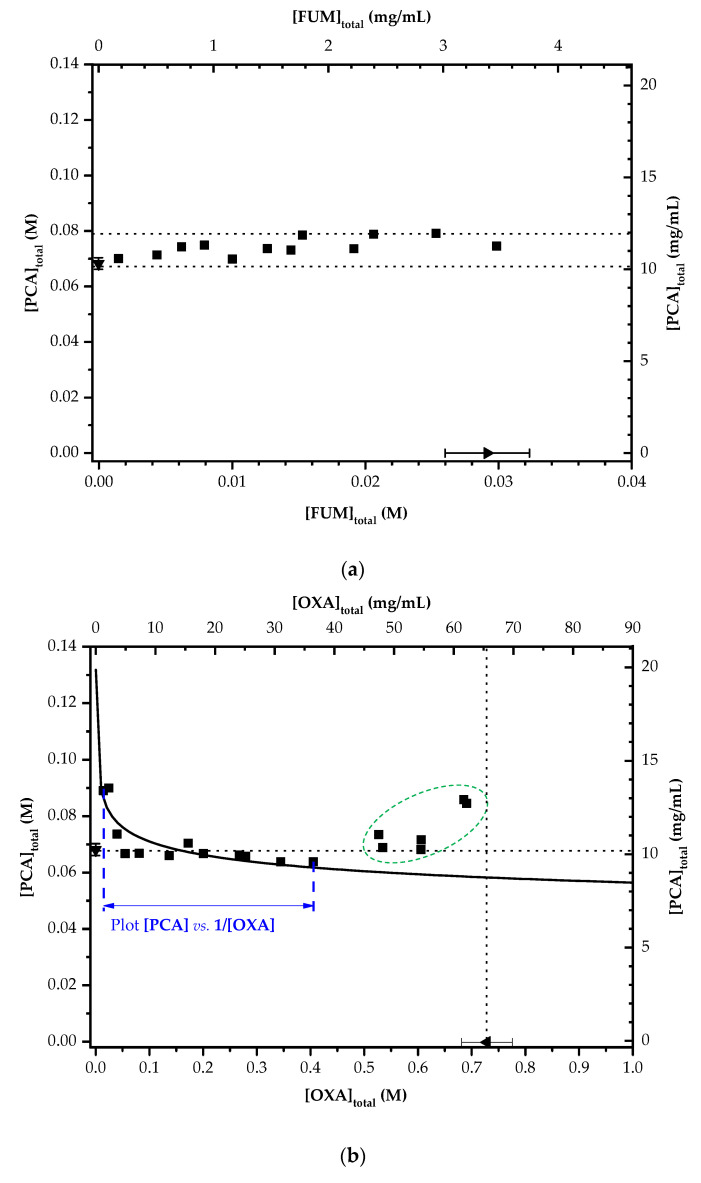
Solubility diagrams of PCA in the aqueous solutions with various concentrations of (**a**) FUM and (**b**) OXA at 10 °C. The solubility values of PCA in water and aqueous solutions of FUM and OXA are labeled by ▼, ●, and ■, respectively, whereas the solubility values of FUM and OXA in water are labeled by ▶ and ◀, respectively. The solid line in (**b**) is a fitting curve based on Equation (10) where *K*_sp_ and *K*_11_ could be obtained in Figure 9.

**Figure 9 pharmaceutics-14-01099-f009:**
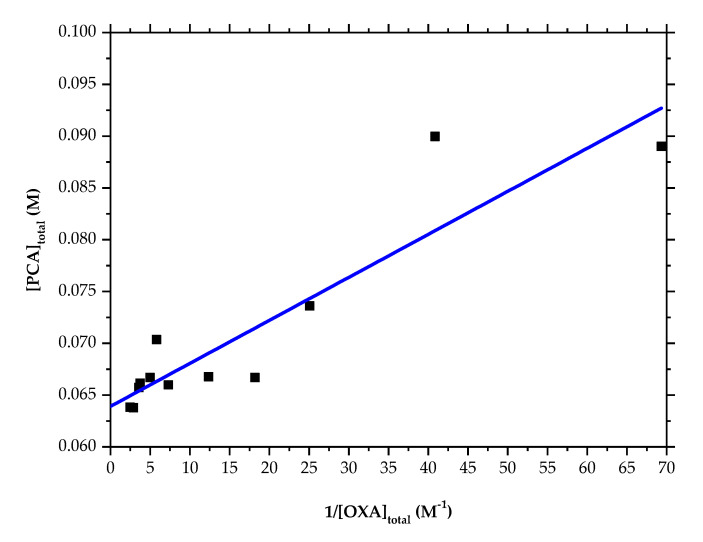
A plot of [PCA] vs. 1/[OXA] for the calculation of *K*_sp_ and *K*_11_.

**Figure 10 pharmaceutics-14-01099-f010:**
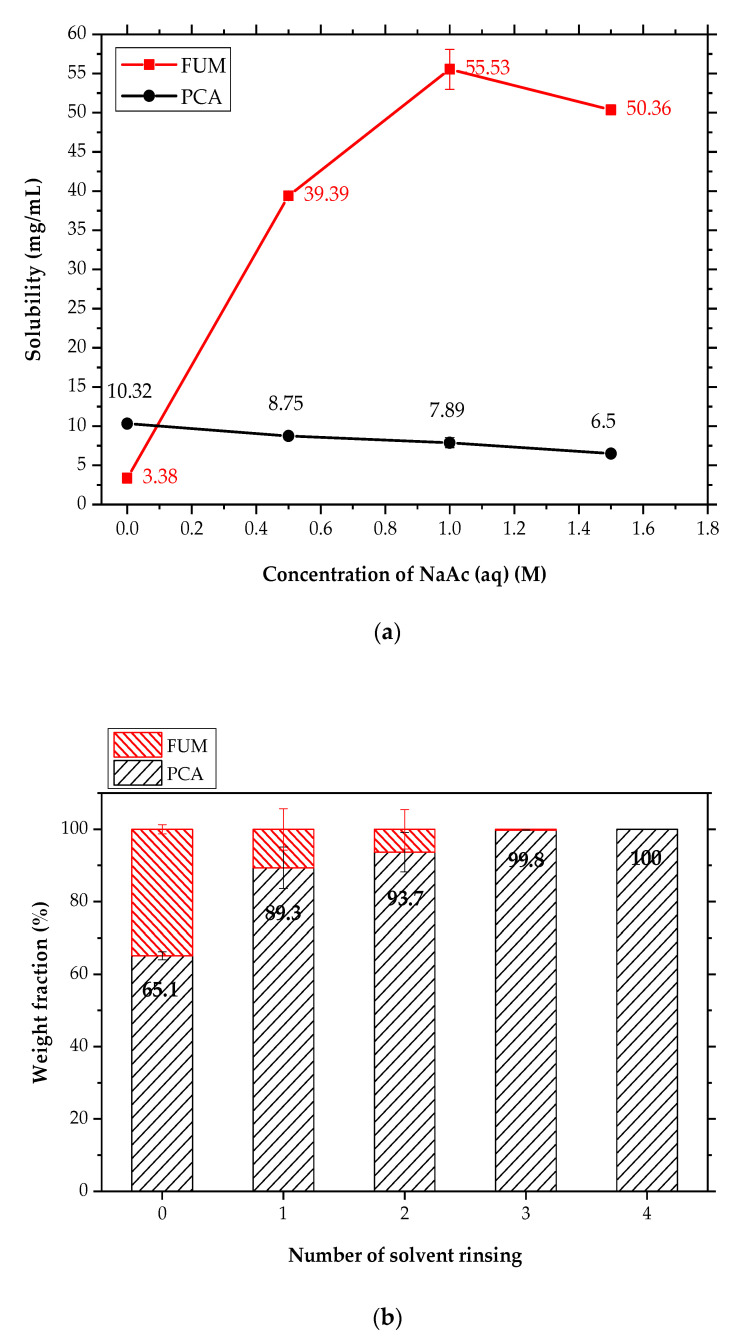
(**a**) Solubility values of Form I PCA (●) and FUM (■) in NaAc (aq) at different concentrations at 10 °C and (**b**) the weight fractions of PCA and FUM in the product on a filter cake after rinsing with 1 M NaAc (aq).

**Figure 11 pharmaceutics-14-01099-f011:**
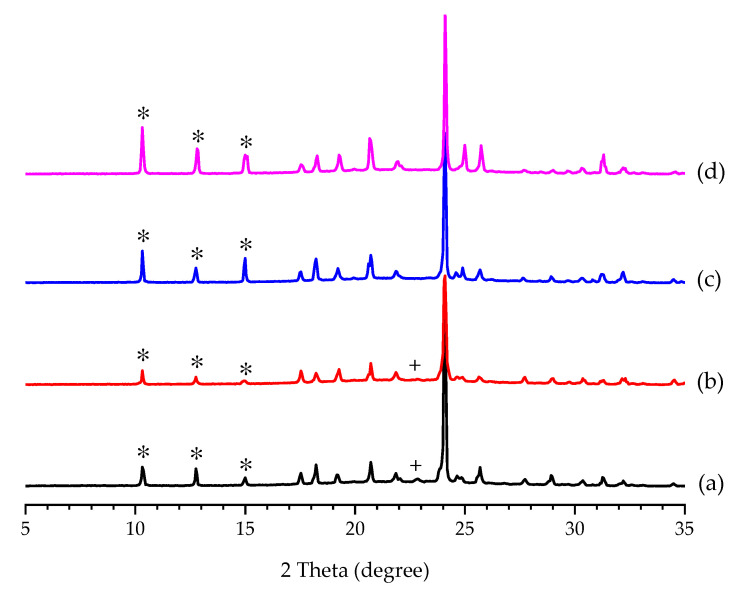
PXRD patterns of (**a**) the PCA crystals produced by cooling crystallization in 50 wt% of FUM and rinsed with 15 mL of 1 M NaAc (aq) at 10 °C (**b**) 2, (**c**) 3, and (**d**) 4 times. The characteristic diffraction peaks of Form II PCA and FUM are labeled by ✴ and +, respectively.

**Figure 12 pharmaceutics-14-01099-f012:**
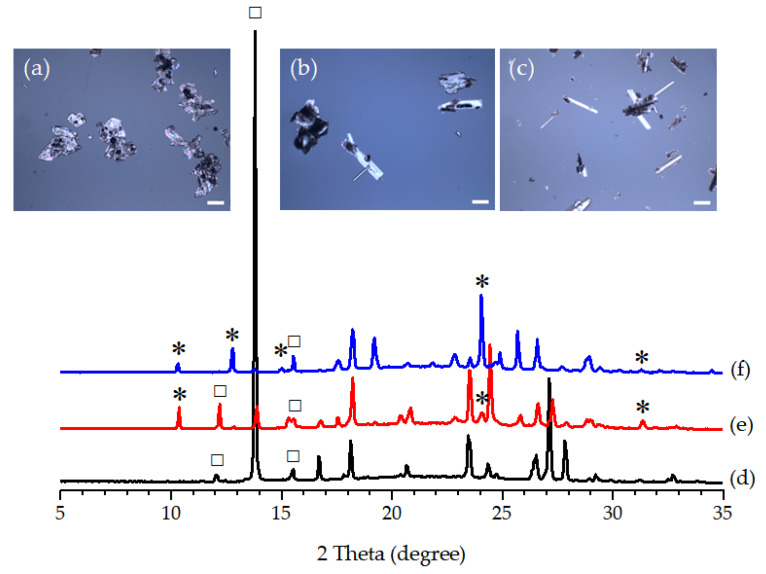
(**a**–**c**) OM images and (**d**–**f**) PXRD patterns of the PCA crystals produced by batch cooling crystallization (**a**,**d**) without an additive, and at (**b**,**e**) 20 wt% and (**c**,**f**) 50 wt% of FUM under no agitation in the 0.5 L vessel (scale bar = 200 μm). The characteristic diffraction peaks of Form I PCA, Form II PCA, and FUM are labeled by ☐, ✴, and +, respectively.

**Table 1 pharmaceutics-14-01099-t001:** Composition of the products produced by Screening Methods 1 and 2 based on PXRD.

Additive	Screening Method 1	Screening Method 2
ADI	Form II PCA + ADI	Form I PCA + ADI
CAF	No crystal	PCA-CAF
CIT	Form I PCA	Form I PCA
FUM	Form II PCA + FUM	Form II PCA + FUM
GLU	Form I PCA	Form I PCA
MAL	Form I PCA	Form II PCA
MAO	Form I PCA	Form I PCA
MLC	Form I PCA	Form II PCA
OXA	Form I PCA	Form II PCA
SUC	Form II PCA	Form II PCA
THP	PCA-THP + Form II THP	PCA-THP + Form II THP
TAR	Form I PCA	Form I PCA

**Table 2 pharmaceutics-14-01099-t002:** Composition of the products harvested by the cooling crystallization of PCA with the assistance of additives at different weight ratios of PCA to additives based on PXRD.

Additive	Weight Ratio of PCA to Additive	Weight Percentage of Additive (wt%)	Composition
ADI	1:0.25	25	Form I PCA
	1:0.5	50	Form II PCA + ADI
	1:0.75	75	Forms I + II PCA + ADI
	1:1	100	Forms I + II PCA + ADI
FUM	1:0.1	10	Form I PCA
	1:0.2	20	Form II PCA + FUM
	1:0.3	30	Form II PCA + FUM
	1:0.5	50	Form II PCA + FUM
MLC	1:0.25	25	Form I PCA
	1:0.5	50	Form I PCA
	1:0.75	75	Form I PCA
	1:1	100	Form I PCA
OXA	1:0.3	30	Form I PCA
	1:0.6	60	Form I PCA
	1:0.9	90	Forms I + II PCA
	1:1.2	120	Form II PCA
SUC	1:0.25	25	Form I PCA
	1:0.5	50	Form I PCA
	1:0.75	75	Forms I + II PCA
	1:1	100	Form II PCA

**Table 3 pharmaceutics-14-01099-t003:** Composition of the PCA products by batch cooling crystallization using the 0.5 L vessel in Expt. 1 to 20.

Expt.	Additive	Weight Percent (%)	Agitation Speed (rpm)	Induction Temperature (°C)	PCA Yield (%)	Composition
1	-	-	300	52 ± 2.7	81.17 ± 0.83	Form I PCA
2	-	-	200	38.3 ± 1.6	80.75 ± 0.74	Form I PCA
3	-	-	100	38.8 ± 1.4	77.94 ± 0.89	Form I PCA
4	FUM	20	300	40.9 ± 2.9	79.47 ± 5.33	Form I PCA + FUM
5	FUM	20	200	41.1 ± 1.8	82.97 ± 1.84	Form I PCA + FUM
6	FUM	20	100	33.2 ± 4.3	77.91 ± 3.01	Form I PCA + FUM
7	FUM	50	300	43.7 ± 4.5	79.06 ± 2.26	Form I PCA + FUM
8	FUM	50	200	38.8 ± 6.4	80.76 ± 1.71	Form I PCA + FUM
9	FUM	50	100	39.2 ± 4.3	78.47 ± 3.32	Form I PCA + FUM
10	OXA	60	300	34.9 ± 6.2	70.94 ± 2.85	Form I PCA
11	OXA	60	200	33.5 ± 6.8	71.94 ± 2.01	Form I PCA
12	OXA	60	100	27.8 ± 9.1	70.23 ± 3.73	Form I PCA
13	OXA	120	300	23.8 ± 11.3	60.71 ± 17.29	Form I PCA
14	OXA	120	200	17.8 ± 0.8	58.06 ± 5.39	Form I PCA
15	OXA	120	100	19.8 ± 3.5	57.48 ± 3.11	Form I PCA
16	-	-	-	33.9 ± 5.3	73.74 ± 4.51	Form I PCA
17	FUM	20	-	18.2 ± 5.7	66.64 ± 5.46	Forms I + II PCA + FUM
18	FUM	50	-	28.2 ± 2.8	72.26 ± 3.72	Forms I + II PCA + FUM
19	OXA	60	-	11.2 ± 1.1	21.21 ± 7.62	Form I PCA
20	OXA	120	-	11.2 ± 1.2	1.91 ± 1.40	Form I PCA

**Table 4 pharmaceutics-14-01099-t004:** Composition of the PCA products by continuous cooling crystallization using the tubular crystallizer in Expt. 21 to 28.

Expt.	PCA (g)	FUM (wt%)	Flow Rate (mL/min)	Yield (%)	Product Composition	
					Collected from Outlet	Remaining in the Crystallizer
21	20	20	75	1.4	Form I PCA	Form I PCA + FUM
22	20	20	150	0.15	Forms I + II PCA	Form I + II PCA + FUM
23	20	30	˙75	0.35	Forms I + II PCA + FUM	Form I + II PCA + FUM
24	20	30	150	0.4	Forms I + II PCA + FUM	Form I + II PCA + FUM
25	20	50	75	-	Clogging	Form I PCA + FUM
26	20	50	150	0.15	Form II PCA + FUM	Form II PCA + FUM
27	15	50	75	4.27	Form I PCA + FUM	Forms I + II PCA + FUM
28	15	50	150	3.13	Form II PCA + FUM	Form II PCA + FUM

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
