# Peer review of "Crystallization of Form II Paracetamol with the Assistance of Carboxylic Acids toward Batch and Continuous Processes"

_pharmaceutics, 2022, doi:10.3390/pharmaceutics14051099_

Round 1

Reviewer 1 Report

The manuscript contains a systematic study for the preparation of paracetamol Form II as a continuation of a previous work (CrystEngComm  2021, 23, 3940). For the initial screening, twelve different coformers were used. Different experimental conditions using seeds in PCA solutions or PCA-additive solutions were employed to finally select the PCA-OXA and PCA-FUM systems. Removal of FUM from the precipitate could be accomplished by solvent rinsing. Finally, scale-up in a stirred tank or tubular crystallizer were studied. Although still some problems must be solved, the latter has resulted a promising option to be considered. Thus, this work is of interest for the readers as it would open new opportunities for other systems.

The manuscript is entitled ‘Batch and continuous crystallization of Form II paracetamol through the assistance of carboxylic acids’.  However, by batch crystallization the objective was not accomplished. It is suggested to look for a more appropriate title which covers all the content of the work.

Have the authors tried to cocrystallize and characterize PCA-fumaric acid as a cocrystal? In the case of PCA-OXA, in page 36 the authors propose the existence of a PCA-OXA2 complex? What about a PCA2-OXA cocrystal? There are many other cocrystals with this ratio 2:1.  In the same was as for fumaric acid, have the authors tried to isolate and characterize the proposed complex?

Although not mandatory, the journal encouraged authors to use the Microsoft word template. Analytical methods and instruments are divided into the manuscript and the SI.  NMR and FTIR descriptions could be also included in the manuscript or move all these techniques to the SI.

In page 16, it is written section 2.2.5 but there is no numbering. Please number sections following the instructions for authors.

It is supposed that Figure 5 should be mentioned in page 30 but it does not appear. However, in page 33, first paragraph, it is written, when it should be figure 6.   Please revise the assignments for figures and table in the document.

Please in the supplementary information describe the extra info as it appears in the SI document: Table S1: Summary of the weights….; Table S2. Experimental….

Reviewer 2 Report

This would be an interesting survey regarding the field of polymorphism and co-crystallization.
Please consider the following

The main purpose of this study is polymorphism formation. Is there any possibility of co-crystal formation?

Is it necessary to perform X-ray single crystal structure analysis?
Please challenge us, as we have obtained beautiful crystalline materials.

Please explain again why you were able to obtain results that are selective for oxalic acid and fumaric acid among the carboxylic acids selected in this study.

In what areas of formulation science do you expect the results of this study to be applied?

Please be sure to provide the structures of the carboxylic acids used. Also, please add the structure of parasetamol.

Thanks,

Reviewer 3 Report

My only query is the extension of Conclusions, but I can accept the present version as it is.

Round 2

Reviewer 2 Report

We see a good comment response and correction.
I feel it would be better to recheck English grammars before the galley stage, if possible.

Thank you for geart.